# Thalamic bursts modulate cortical synchrony locally to switch between states of global functional connectivity in a cognitive task

Oscar Portoles[1,2]*, Manuel Blesa[3], Marieke van Vugt[1], Ming Cao[2], Jelmer P. Borst[1]*

**1** Bernoulli Institute for Mathematics, Computer Science and Artificial Intelligence, Faculty of Science and Engineering, University of Groningen, Groningen, The Netherlands, **2** Engineering and Technology Institute Groningen, Faculty of Science and Engineering, University of Groningen, Groningen, The Netherlands, **3** MRC Centre for Reproductive Health, University of Edinburgh, Edinburgh, United Kingdom

* o.portoles.marin@rug.nl (OP); j.p.borst@rug.nl (JP)

**Data Availability Statement:** All code written in support of this publication, processed data, simulation input files and simulation out put files are publicly available at https://gin.g-node.org/

## Abstract

Performing a cognitive task requires going through a sequence of functionally diverse stages. Although it is typically assumed that these stages are characterized by distinct states of cortical synchrony that are triggered by sub-cortical events, little reported evidence supports this hypothesis. To test this hypothesis, we first identified cognitive stages in single-trial MEG data of an associative recognition task, showing with a novel method that each stage begins with local modulations of synchrony followed by a state of directed functional connectivity. Second, we developed the first whole-brain model that can simulate cortical synchrony throughout a task. The model suggests that the observed synchrony is caused by thalamocortical bursts at the onset of each stage, targeted at cortical synapses and interacting with the structural anatomical connectivity. These findings confirm that cognitive stages are defined by distinct states of cortical synchrony and explains the network-level mechanisms necessary for reaching stage-dependent synchrony states.

## Author summary

A novel machine-learning method was applied to unveil the dynamics of local and cortex-wide neural coordination underlying the fundamental cognitive processes involved in a memory task. To explain how neural activity–and ultimately behavior–was coordinated throughout the task, we developed a whole-brain model that incorporates cognitive mechanisms, anatomy, and neural biophysics. Similar models are regularly used with resting state data, but simulating a cognitive task remained elusive. By using hidden semi-Markov models to divide the task into stages with separate connectivity patterns, we were able to generalize the whole brain model from resting state to cognitive task data. The model showed that sub-cortical pulses at the onset of cognitive processes–as hypothesized by cognitive and neurophysiological theories–were sufficient to switch between the states of neural coordination observed. These findings have implications for understanding goal-

oportoles/SwitchingFCalongAtask. Raw data is available at https://www.jelmerborst.nl/models/.

**Funding:** MC was supported in part by the European Research Council (ERC-CoG-771687); https://erc.europa.eu/. The PhD of OP was largely funded by the Data and Systems Complexity Centre of the University of Groningen. The funders had no role in study design, data collection and analysis, decision to publish, or preparation of the manuscript.

**Competing interests:** The authors have declared that no competing interests exist.

directed cognitive processing and the mechanisms needed to reach states of neural coordination.

## Introduction

Already in the 19<sup>th</sup> century, Donders hypothesized that information processing in the brain proceeds through a sequence of fundamental cognitive stages with different functions such as visual encoding, memory retrieval, and decision making [1]. Initially, cognitive stages were investigated with behavioral metrics like reaction time (e.g., [2]). Over the past decade, neuro-imaging analyses have begun to uncover the neural correlates of these cognitive stages (e.g., [3]).

The dominant view is that cognitive stages require specific patterns of neural coordination across the cortex [3–5]. The transition from one cognitive stage to the next is thought to be driven by the basal-ganglia-thalamus (BGT) system which sets new states of cortical coordination [6–8]. The striatum monitors the current state of the cortex, and based on a comparison to predefined states, selects and triggers the next cognitive stage. The role of the BGT system modulating cortical coordination is supported by animal studies, intracranial recordings, and neural models [9–14]. However, the network-level mechanisms required to reach a new state of cortical coordination from subcortical inputs are poorly understood.

To give a detailed account of these mechanisms, one first needs to characterize the different states of neural coordination within the sequence of cognitive stages. We measured neural activity with cortically-projected magnetoencephalographic (MEG) recordings as these have a sufficiently fine temporal resolution to measure cognitive stages, as well as adequate spatial resolution [5]. However, cognitive stages have high temporal variability–that is, stages typically have a different duration on each trial of an experimental task–which makes it difficult to measure neural coordination. To overcome this problem, we used a machine learning method that identifies the onsets of cognitive stages on a trial-by-trial basis [15]. Afterwards, the identified stage onsets were used to time-lock the measures of neural coordination within regions (local synchrony) and between regions (functional connectivity, FC), as there are concurrent changes at both spatial scales [16,17].

Specifically, we focused on coordination of theta band oscillations as the thalamus is thought to modulate local theta activity [18] which may change theta band FC [19], and which in turn may modulate the activity in higher frequency bands [20].

The machine learning method that we used to identify cognitive stages combines multivariate pattern analysis with a Hidden semi-Markov Model (HSMM-MVPA). The HSMM-MVPA method searches in each trial for a sequence of short-lived modulations of MEG amplitude (hereafter called *bumps*, following the original paper [15]) that have a consistent topology across trials. These bumps signify the onset of cognitive stages, and are thought to be triggered by the BGT system. Previously this method has been used successfully to, for example, identify the cognitive stages that are affected by task manipulations such as difficulty, stage insertion, and evidence accumulation for decisions [4,5,15,21].

To understand how events from the BGT system can cause switches between states of neural coordination–and thus between cognitive stages–we build upon generative whole-brain biophysical models of large-scale activity (GWBM) that have been used to explain the dynamics of neural coordination at rest [22]. GWBMs reduce the whole-brain network of neurons and synapses to a smaller network that still incorporates the most relevant principles of neural

dynamics. The nodes of such a network describe the macroscopic activity within a region, while the links reflect the neural fibers that connect these regions (i.e. structural connectivity).

GWBMs of resting state indicate that time-resolved patterns of neural coordination depend on the anatomical structure of the brain and that these patterns evolve without requiring any input (a phenomenon referred to as *metastable coordination*; [17,22]. Such coordination dynamics are thought to provide an optimal mechanism for simultaneously integrating and segregating information that allows the system to adapt quickly or alternatively, to persist in a given state [23]. While this is sufficient to explain resting-state data, cognitive tasks require specific, controlled sequences of coordination states.

Here, we explored a GWBM in which inputs from the BGT system modulated local connectivity strength briefly at the onset of cognitive stages, as suggested by cognitive theories and electrophysiology measurements [6–14]. In other complex networks with similar dynamics as the brain, such local perturbations can, in turn, produce controlled switches between global states [24]. Similarly, even though the inputs from the BGT system only triggered direct changes in local connectivity strength, we observed transient modulations of local synchrony and switches to the targeted states of directed functional connectivity that lasted until the next input. When there were no further inputs from the BGT system, neural coordination returned to resting-state patterns after tens of seconds. These results matched the observed neural coordination throughout the cognitive stages in the empirical data. Finally, we used the GWBM to determine the importance of each brain region in facilitating the switches between states of coordination.

## Results

### Five cognitive stages in an associative memory task

We re-analyzed MEG data from an associative memory recognition task with 18 participants [3]. We chose this task because associative recognition memory involves a rich variety in cognitive stages that have also been widely studied [3,5,15,25,26]. The task consisted of a self-directed learning phase during which participants memorized 32 word pairs and a test phase. In the test phase–which we analyzed here–participants were again presented with word pairs. These could be *target* pairs from the learning phase or *re-paired foil* pairs, which consisted of the same words paired differently (e.g., if the participants learned apple-tree and month-house, a foil pair could be apple-house). Participants were asked to indicate as quickly and accurately as possible with a key press if it was a learned pair or a re-paired foil. Only correct responses were included in our analysis. We were interested in the evolution of neural coordination along with the cognitive stages involved in performing the task, and in particular in how the brain switches between these consecutive states of functional neural coordination.

As the goal is to develop a cortical model, the MEG signals were projected onto 5,124 cortical sources using the structural MRI of each participant with minimum-norm estimation [3]. The resulting cortical activity was aggregated into 68 cortical regions following the Desikan-Killiany atlas [27]. Each cortical region in the atlas contained the average activity of the cortical sources within that region. Next, HSMM-MVPA was used to estimate the timing of bumps that indicate the onset of cognitive stages in each trial. All trials were assumed to go through the same sequence of stages as in previous studies [3,4,15]. Thus, bumps were assumed to have the same spatial topology across trials, but trial-to-trial variable temporal location. Nevertheless, the HSMM-MVPA can cope relatively well with extra bumps in some trials [15]. The intervals between stimulus-onset-to-bump, bump-to-bump, and bump-to-response constitute the cognitive stages. A leave-one-subject-out cross validation method showed that the MEG

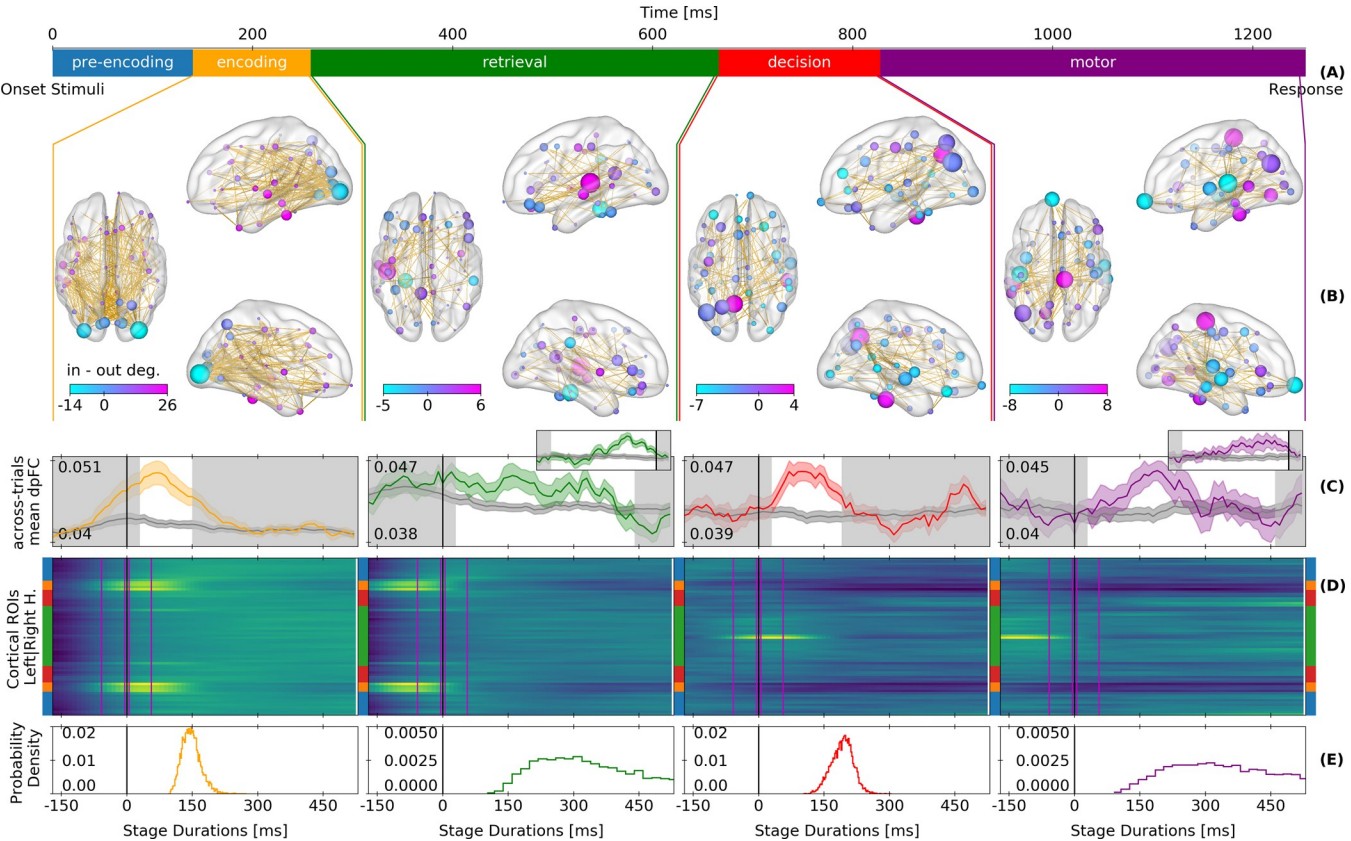

**Fig 1. Theta-band MEG local synchrony and directed functional connectivity by cognitive stages.** (A) Cognitive stages derived with the HSMM-MVPA along with their median durations. (B) Significant directed functional connectivity throughout the stage (*within-stage dpFC*). Links go from phase-ahead to phase-behind regions. The nodes represent the nodal degree (size) and the difference between phase-ahead and phase-behind links (color). (C) Directed functional connectivity at every sample time-locked at the onset of the stages (*across-trials dpFC*). Colored (dark gray) line: average across links with (without) significant across-trial dpFC at the current stage; Shading: standard error of the mean across subjects. Black vertical lines indicate the onsets of the stages. The white background spans the median stage duration. Retrieval and response insets: Directed functional connectivity time-locked to the onsets of the decision stage and to the end-of-trial response, respectively (D) Across-trials averaged local synchrony (z-scored envelope of amplitudes) time-locked at the onset of the stages. Y-axis represents cortical regions – blue: temporal, orange: occipital, red: parietal, and green: frontal. Magenta lines define the time windows used to measure the relative change in local coordination at the onset of the stages. First and second windows span -60 to -10 ms, and 0 to 50 ms with respect to stage onset. (E) Histogram of stage durations derived with the HSMM-MVPA.

data were best explained by a HSMM-MVPA model with four bumps, which corresponds to five cognitive stages (Fig 1A).

Following previous work on associative recognition [3,5,15], we interpreted the five cognitive stages as follows: pre-encoding, encoding of visual information, memory retrieval, decision making, and motor response. We did not analyze the pre-encoding stage as it is mostly driven by the task stimulus and not by events from the BGT system that produce the transitions between cognitive stages. The retrieval and motor stages were longer and had larger across-trial variability than the encoding and decision stages (Fig 1A and 1E).

## Different local synchrony and directed functional connectivity states between stages

Next, we measured neural coordination in the discovered stages. We focused on coordination of theta band oscillations (4–8 Hz), for several reasons: we previously found synchrony patterns in this frequency band to vary across task stages [4]; theta oscillations have been related

to cognitive processes such as attention, memory, control, and decision making [28–31]; the phase of theta oscillations is known to modulate the activity in higher frequency bands [20,29]; local modulations of theta-band activity are hypothesized to mediate changes in long-range functional connectivity [19]; and thalamic activity modulates cortical theta band activity [18].

Directed FC in the theta band was operationalized by means of the directed phase-lag index [32] (dpFC). The directed phase-lag index measures the consistency of the sign of the difference between the phases of two signals. Such consistency can exist either over a period of time or across trials at a given time point. We measured first *within-stage dpFC* to capture directed FC states that are constant from the start to the end of a cognitive stage. Fig 1B shows the links with significant within-stage dpFC, as well as the local difference between phase-ahead and phase-behind links (node color) and the total number of links regardless of their direction (node size). The significance of dpFC was obtained using a permutation in which we created 200 surrogate data sets with random circular shifts of the original phases. Such circular shifts keep the structure of the phases, while they destroy the temporal relationship between a pairs of phase signals. We then examined whether the empirically observed phase lags were more consistent than this population of dpFCs from randomly shifted signals. A significant dpFC indicates that the phase in one region is consistently ahead or behind another region. Next, we measured *across-trial dpFC* to reveal, sample-by-sample, the temporal evolution of sign-consistent phase differences across trials during each stage (Fig 1C). Across-trial dpFC was calculated at every sample with the trials time-locked to the onset of each of the stages. Time-locking to the onset of stages rather than an absolute time relative to stimulus onset allows for aligning cognitive processes that start at different times on each trial.

Across-trial dpFC revealed that functional states of directed FC switch at the transition between cognitive stages. These switches are visible because across-trial dpFC takes into account the trial-to-trial temporal variability of the cognitive stages as revealed by the HSMM-MVPA analysis. For example, for the memory retrieval stage, across-trial dpFC seems to fade halfway through. However, when across-trial dpFC is time-locked to the onset of the next stage–the decision stage–dpFC for memory retrieval materializes until shortly before the decision stage (see the insets in Fig 1). This illustrates why the HSMM-MVPA analysis is crucial: otherwise dpFC would appear to fade quickly after stimulus onset, while that is not the case when first isolating cognitive stages.

Local synchrony was operationalized as the envelope of the theta band analytic signals in each region, which indicates the degree of synchronous neural activity within a region. The envelopes were z-scored over time and then averaged across trials and participants. Across-trial averages were time-locked to the onset of cognitive stages which gave a time course of local synchrony for each stage (Fig 1D). This showed that the local modulations of synchrony occurred only briefly at the start of each stage, and involved different regions depending on the cognitive operations involved in that stage.

As expected, each stage had a different neural coordination pattern. In the visual encoding stage, occipital and left-temporal regions showed local synchrony and dpFC which might facilitate the transfer of visual information to the medial temporal lobe and the hippocampus to start a retrieval process [26]. The encoding of information is controlled by a large fronto-posterior, fronto-lateral network [28,29]. During the memory retrieval stage, local synchrony at occipital and temporal regions is reduced. dpFC now happens mostly between left-medial-temporal and frontal regions, whose coordination is required for memory tasks [26,33]. At the onset of decision making the frontal regions begin to synchronize locally. The decision process is mediated by fronto-parietal dpFC [30], and dpFC between temporal and parietal regions dpFC to reinsert the memory retrieved into the left-parietal cortex [26]. Finally, at the motor stage a large dpFC network appears between motor, temporal, left-parietal, and pre-frontal

regions. This complex network has previously been associated with motor preparation, action reevaluation, decision, and cognitive control [30,31,34], in line with the idea that the action is reevaluated during the motor response [35].

Together, these analyses unveiled that right at the onset of a cognitive stage there is a reorganization of neural coordination in the cortex. Whereas the change in local synchrony was only brief, dpFC lasted throughout the cognitive stage, indicating that short modulations of local synchrony can have persistent global effects. Next, we used a GWBM to investigate the mechanism underlying this.

## Generative large-scale whole-brain model (GWBM)

In order to integrate cognitive stages and neural coordination into one framework along with neural anatomy and neural dynamics, we used a parsimonious GWBM that describes within- and between-region modulations of synchrony. Previously, we have used this model to demonstrate that modulations of local synchrony are related to time-resolved FC during resting state [17].

This GWBM is a low-dimensional reduction of a network-of-networks of Kuramoto oscillators [36]. Kuramoto oscillators describe the dynamics of synchrony in biological systems including neural networks [22,37,38]. Each sub-network in this study represents a cortical region from the Desikan-Killiany atlas. All units in a region are assumed to be fully and instantly connected, while connections between regions are weighted and delayed by the density and length of the neural fibers in MRI-derived structural connectivity networks. The regions in the GWBM were defined with the same parcellation atlas as the MEG data that we sought to model.

First, we set default values for local connectivity strength ($L$ in Eq 3, identical for all regions), and global scaling ($G$ in Eq 3 and Eq 4) of the structural connectivity such that the model simulated resting-state coordination dynamics in the theta-band [22,39]. Resting-state dynamics are characterized by fluctuations over time of the local and global synchrony as well as time-resolved FC patterns (i.e. local and global metastability) [17,22,39]. These dynamical properties of resting state neural coordination were identified with GWBMs simulated over a grid of $L$ and $G$ values. The identified $L$ and $G$ values displayed the most similar dynamics to local and global metastability in the grid search. (see S1A Fig).

Next, we simulated the switching between cognitive states by adding short inputs (30 milliseconds) from the BGT system–specifically from the thalamus to the cortex–at the onset of cognitive stages. The rationale for using this mechanism derives from theories of cognition and data derived from electrophysiology. Specifically, cognitive theories state that the BGT system modulates cortical synchrony at the onset of cognitive stages via thalamocortical signals [6,25]. Electrophysiology has shown that thalamocortical neurons can indeed drive cortical activity [10,13] and establish FC [40,41]. These thalamocortical neurons tend to produce short burst of activity [42], which target pools of either excitatory or inhibitory cortical neurons specifically [12,43]. Therefore, our model simulated thalamocortical inputs as short pulses of increased or decreased local connectivity strength ($L$ in Eq 3) that represent transient modulations of excitatory or inhibitory synaptic activity [38].

To simulate the sequence of neural coordination states found in the MEG data, we estimated the magnitude of the required activity pulses simultaneously in all regions, stage-by-stage. Each cortical region received one pulse at the onset of each processing stage. The optimization scheme to obtain the magnitude of the pulses maximized concurrently the fitness of local synchrony and within-stage dpFC, while minimizing the total magnitude of the pulses.

The optimization was accomplished with the generalized island model for distributed evolutionary optimization which in relatively short time explores and exploits different areas of the parameter space simultaneously [44].

## Changes in local connectivity cause switches between global states of cognitive coordination

To assess how well the model simulated local synchrony, we measured the relative change in theta envelope before and after stage onset (magenta lines in Fig 1D). All model results were computed from 1,000 models randomly selected from the top one percentile of models after the optimization. We used 1,000 GWBMs to derive our conclusions in order to obtain the general behavior of the GWBM and not the behavior of a single parametrization of the model that could reflect local optima. Relative changes in simulated and MEG envelopes were correlated significantly across different cortical regions (Spearman's $\rho$–encoding: 0.552 ± 0.00158 SEM; retrieval: 0.702 ± 0.000434 SEM; decision: 0.743 ± 0.000683 SEM; motor 0.477 ± 0.00151 SEM; all p-values < 0.05).

Fig 2A compares the within-stage dpFC fitness of the *worst* model in the top one percentile to a distribution of the same fitness metric obtained with 20,000 random within-stage dpFCs, and shows that the model performs much better than chance. Fig 2B shows the fitness of within-stage dpFC at individual links. The fitness was quantified as the proportion of links with the same phase-lag direction as in the MEG within-stage dpFC (encoding: 0.697 ± 0.00014 SEM; retrieval: 0.837 ± 0.0016 SEM; decision: 0.749 ± 0.00092 SEM; motor: 0.758 ± 0.001 SEM). Fig 2C compares the across-trial dpFC of the model to the MEG data over time. Each state of dpFC begins after the pulse that modulates local connectivity strength at the onset of the stage, and vanishes with the next onset (Fig 2C). The last state of dpFC–the

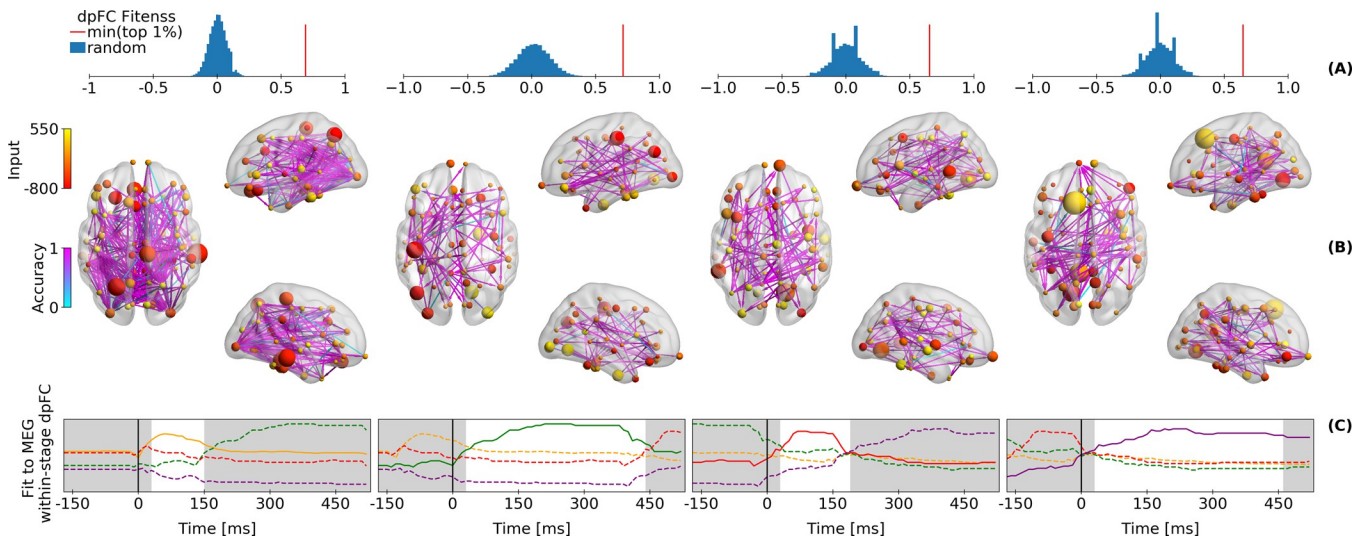

**Fig 2. Simulated directed functional connectivity.** (A) Blue histograms show the fitness between 20,000 randomly generated *within-stage dpFCs* and the MEG *within-stage dpFC*. The red line indicates the *within-stage dpFCs* fitness of the model with the lowest fitness index within the top 1 percentile of the optimized models. (B) Fitness of simulated-to-MEG *within-stage dpFC* is shown in cyan-purple grading over MEG links with significant *within-stage dpFC* (same as Fig 1B). The nodes indicate the relevance of a region for reaching a state of within-stage dpFC (size), and the pulse of local connectivity strength at the onset of the stage due to sub-cortical inputs (colors). See S4 Fig for the standard error of the means. These results show the averages of 1000 random picks from top ~1% of the optimizations with the best fitness index. (C) Temporal evolution of simulated-to-MEG fitness of *within-stage dpFC* for the current stages (solid lines) compared to other stages (dashed). The white background spans the median stage duration. The colors of the lines represent the different stages and follows Fig 1C.

motor response–vanishes slowly (in ~10 seconds), and the GWBM returns to resting-state coordination dynamics (S2 Fig).

Taken together, the GWBM showed that a short pulse of local connectivity strength at the onset of a cognitive stage can first cause a modulation of local synchrony and then initiate a new state of dpFC that lasts until the onset of the next stage (Fig 2C). If there is no subsequent cognitive stage, the GWBM returns to the coordination dynamics that are characteristic of the resting state.

### Relevance of regions to switch between functional states of coordination

Not all regions in the GWBM are equally important for switching between states of dpFC. The relevance of a region increases with the size of the pulses and the strength of structural connectivity with other regions. The size of the nodes in Fig 2B indicates the relevance of a region for switching between states of dpFC. The relevance of each node is the average of the 1,000 GWBMs randomly picked from the best-fitting ~1% GWBMs in the optimization process. Although each of the 1,000 GWBM had slightly different parameters (S3 Fig), the GWBMs had similar dynamics and gave consistent results as the small SEM show here and in S4 Fig. The absolute size of the pulses from the BGT predicts 22.52% (± 0.096 SEM) of the variance in the relevance of the nodes, while the interaction between the absolute size of these BGT pulses and the log-scaled strength of structural connectivity predicts 25.98% (± 0.12 SEM) of the same variance.

This analysis shows that there are regions such as the left superior frontal region in the last stage that do not show dpFC, but that are still highly relevant for entering a state of high dpFC between other regions. This supports the mechanistic role of the superior frontal regions in exerting cognitive control [28,30] and highlights the complexity of interactions required for implementing changes in FC patterns.

## Discussion

In this paper, we first analyzed the evolution of macroscopic neural coordination states across the cortex during an associative recognition memory task. Our analysis of MEG data showed that at the onset of fundamental cognitive stages there are transient modulations of local synchrony, which are directly followed by a new state of dpFC that persists until the next cognitive stage. Next, we used a generative model of whole brain activity (GWBM) with inputs from the basal-ganglia-thalamus system to explain these findings. The GWBM showed that short pulses that strengthen or weaken local connectivity strength at the onset of cognitive stages were sufficient to cause the switch between states of neural coordination consistent with empirical data. In addition, the GWBM indicated which individual regions were most relevant for causing these switches.

### GWBM and the Basal Ganglia-Thalamus Circuit

The GWBM that we developed in this paper has shown that inputs from the basal-ganglia-thalamus system at the onset of cognitive stages are sufficient to cause switches between cognitive stages. This role of the BGT system had been hypothesized by cognitive theories [6,25]. Direct evidence for thalamic modulations of cortical activity is limited to some cortical regions due to methodological constraints, such as the fact that it is challenging to record simultaneously from many sub-cortical and cortical areas with high temporal resolution [10,12,13,40]. Nevertheless, a recent meta-analysis has shown that the thalamus plays a critical role as a central hub that connects neighboring and distant regions to allow for cognitive functions [9], which is in line with the hypothesis that local thalamocortical inputs can mediate FC [14]. In addition,

there is evidence for neural fibers connecting the thalamus with most cortical regions [45]. Our model provides additional support for both hypotheses: the BGT systems can trigger a switch between fundamental stages of cognition [6,25] and thalamic input modulates coordination of cortical activity according to cognitive demands [14].

Given our limited understanding of how the thalamus modulates cortical activity, we opted for a very simple representation of thalamocortical input. These inputs were short [40–42], targeted excitatory or inhibitory local connections [12,43], and came at the onset of cognitive stages [6,25]. Such inputs drove the GWBM throughout the sequence of empirical local synchrony and dpFC states. Afterwards, the GWBM returned to resting state dynamics. In other words, a short modulation of local excitation/inhibition modified the local synchrony and the phase of the local mean-field oscillation. This change in local mean-field phase set a new phase-lag relationship with other regions that vanished over time due to structural cortical interactions. This response to local perturbations suggests that cortical dynamics are metastable as many states of coordination can be reached, and the brain does not remain into a particular state in the absence of perturbations. Metastable dynamics are thought to allow for integrating and segregating information simultaneously, as well as for the flexibility of cognitive functions and behaviors [23].

Importantly, dpFC was not driven by thalamocortical inputs exclusively. Instead, the macroscopic connectivity structure of the brain also played an important role. The importance of the structural connectivity was highlighted by the presence of regions with very low dpFC that turn out to be very important for coordinating other pairs of regions as the analysis of the relevance of single regions for switching between states of dpFC shows. One example of such regions is the left superior frontal region during the motor response stage, a region that has been related to cognitive control, attention, and decision making [28,30,31]. The role of structural connectivity in generating specific coordination patterns was first brought to light by GWBMs of resting-state dynamics [22]. In a previous study we have shown analytically that the strength of structural connectivity plays an important role in selectively coordinating regions by means of modulations of local connectivity strength [46]. Additionally, structural symmetries and time delays might have influenced dpFC in our simulations [22,47,48].

There are other biological aspects that might be relevant for coordination of cortical activity that were not included here, including the delay over thalamocortical neurons [49], the dynamics of the synapses targeted by thalamocortical inputs [12], tonic activity in the thalamus [42], noise, or the state of cortical oscillations at the time of a thalamic input. Moreover, our measurement of directed FC has neglected zero-phase-lag coordination which can emerge from thalamocortical and cortico-cortical loops [49]. However, while including these additional aspects might improve the fit of the model, the current model could already account for the data surprisingly well. Additionally, we have assumed that perturbations of cortical dynamics at the onset of cognitive stages come exclusively from the thalamus. However, there might be other regions such as the hypothalamus that modulate cortical activity in the same or another way that is not included in our model.

This is the first GWBM that can simulate local and cortex-wide neural coordination throughout a cognitive task. In addition to modeling the transition between cognitive stages, this study overcame several difficulties to make it feasible to simulate different neural coordination states occurring over the course of a cognitive task, some of which may be applied to simulating resting state as well. Mainly, we developed a method to derived the initial conditions of the GWBM (phases and amplitudes) at the beginning of a task based on the state of neural coordination at that time. Second, we defined a fitness function that incorporates local and cortex-wide synchrony and focuses on relevant properties of the data (i.e., significant dpFC and modulations of local synchrony). Third, the optimization algorithm in combination

with the fitness function made computationally feasible to reliably estimate $68^*4+1$ parameters.

## Neural coordination across the cortex along a sequence of cognitive stages

Our novel approach to measuring local synchrony and dpFC time-locked to the onset of cognitive stages revealed with high temporal resolution that each cognitive stage has a particular pattern of dpFC. This stage-dependent pattern of dpFC starts at the onset of the cognitive stage and vanishes at the end of the same cognitive stage. Furthermore, a switch between consecutive states of dpFC has modulations of local synchrony in-between both states of dpFC–the onset of the cognitive stage. Our stage-by-stage analyses of neural coordination are consistent with the hypothesis that the local modulations of phase synchrony in the theta-band mark a change in long-range functional connectivity and enable a new cognitive function [19]. Moreover, our results support the hypothesis that a new state of neural coordination is established at the onset of cognitive stages [6,25]. Our previous research has shown that alpha band FC also varies across cognitive stages [4], but cortical alpha has been found to lead thalamic activity rather than being caused by it [50], as is the case with theta [13].

To uncover neural coordination stage-by-stage it was crucial to account for the temporal variability of cognitive stages across trials using the HsMM-MVPA analysis. Only after correcting for this variability, our analyses showed that dpFC lasts throughout a cognitive stage and differs across stages. The corresponding states of dpFC had different length, strength, and topology. This diversity of properties might have biased some traditional metrics of neural coordination. For example, if one were interested in the FC at the interval between 250 and 600 milliseconds after stimulus onset–roughly the period of memory retrieval, this interval would contain elements of the encoding or decision stages. The first reason for this is the trial-by-trial variability in stage durations: in one trial encoding might last till 400 ms, while in another trial memory retrieval might already have finished by 400 ms. Secondly, the retrieval stage has fewer and weaker connections than the encoding and decision stages in our study, which mean that these connections might have been missed altogether. These effects are worse the further one moves away from fixed time points (trial onset/response), which is one of the reasons that M/EEG studies have had severely limited trial lengths traditionally.

Furthermore, our stage-by-stage analysis might contribute to disentangling competing theories. For example, our results suggest that the decision is made and evaluated in the last two stages. We interpreted the penultimate stage as a decision process in which memories are transferred to parietal areas by coordinating left-temporal regions with parietal regions, mediated by local frontal and fronto-parietal coordination [15,26,28,30]. The last stage has been traditionally related with a pure motor response. However, our results indicate that the motor stage has elements associated with motor preparation, action reevaluation, decision, and cognitive control [29,31,34]. This functional network in the last stage suggests that during the motor stage the decision is reevaluated, and it supports the line of thought in which responding is a process that is not independent from decision making (e.g., [31,35]).

## Conclusion

To the best of our knowledge we have developed the first generative large-scale brain model that simulates the dynamics of the states of neural coordination along the fundamental cognitive stages in a task. In this model we have integrated structural connectivity, macroscopic neural dynamics, sub-cortical inputs, and the cognitive theories of associative recognition memory. The model has multiple simplifying assumptions which made it feasible to simulate and optimize the model while taking into account the macroscopic properties of neural

anatomy and dynamics. This work opens up the way for considering other tasks in similarly integrated and multidimensional manners to better understand how the brain implements cognition through cortical coordination.

## Methods

### Experimental paradigm

We re-analyzed MEG data from an associative memory task [3]. We combined the trials with correct responses from all experimental conditions, as we were interested in the transition between fundamental cognitive stages and not in the differences between conditions (which did all proceed through the same stages; [15]). All 18 participants were right-handed (6 males and 12 females with a mean age of 23.6 years).

First, participants studied 32 pairs of words until they knew them well [3]. This was followed by a test session in which MEG was recorded. In the test session participants were presented with pairs of words which were either the same as in the study season (*targets*) or paired differently (*re-paired foils*). The pairs of words remained on the screen until the participant responded, and were followed by 1-sec feedback and a brief inter-trial interval. A full description of the task and the recording procedure can be found in [3].

### MEG data preprocessing

MEG data was preprocessed and source-reconstructed following the analysis pipeline of the original manuscript [3]. After artifact rejection there were 6,708 trials left. The MEG data of each participant was combined with their own structural MRI to obtain the cortical sources of MEG data. MEG sources consisted of 5,124 dipoles estimated with cortically constrained minimum norm estimates [3,51]. Source estimates were then morphed onto the standard MNI brain and parcellated into 68 cortical regions with the Desikan-Killiany atlas [27,52]. Each parcel contained the average activity of all dipoles within the region with a 100 Hz sampling rate.

### Identification of cognitive stages

To find the onset of cognitive stages the data were bandpass filtered (1–30 Hz, which are default values in Field Trip [53]) and epoched from trial onset to response. Single trials were baseline corrected (-0.4 to 0 seconds), and transformed to one covariance matrix per subject. The average covariance matrix across subjects was used to reduce the dimensionality of the data to 33 principal components (which together accounted for 90% of variance). These principal components were z-scored and fed into the HSMM-MVPA. The HSMM-MVPA first applies a half-sine window function to increase the signal-to-noise ratio of the *bumps*, the cortical response to sub-cortical input. The bumps are assumed to be 50-millisecond modulations of amplitude at the onset of cognitive stages with the same topology across trials. The signals from the end of a bump to the next bump are assumed to have zero-mean amplitude, a *flat*. The duration of a given stage (bump + flat) is assumed to come from a gamma distribution with shape parameter equal to two, which turns out to be the most suitable shapes for modeling the durations of cognitive processes [15] Consequently, a stage is modeled as a bump of a certain amplitude followed by a zero-mean amplitude flat and a duration given by a gamma-2 distribution. There is one exception and this is the first stage (pre-visual encoding here) which does not start with a bump. With this stage model and a predefined number of stages, the Baum-Welch algorithm for HSMMs searches the amplitude and location of bumps that explain the z-scored principal components best [54]. The bump amplitudes (for the 33 PCA components) are the same for all trials and vary across stages. The temporal location of the

bumps also varies across trials, but the resulting stage durations are constrained to gamma-2 distributions with one scale parameter per stage.

We explored models with 3 to 7 cognitive stages as previous studies have shown that this memory task consists of 5 to 6 stages [4,5,15]. For a model with $N$ stages we ran the HSMM-MVPA 200 times with random initial parameters to avoid converging in local maxima. To select the most representative number of stages for all subjects we used a leave-one-subject-out cross-validation to obtain the likelihood of fitting the MEG data of a subject not used to train the HSMM-MVPA. Next we used a sign-test to assess whether a HSMM-MVPA with $N+1$ stages could explain the MEG data of more subjects significantly better than a HSMM-MVPA with $N$ stages [15]. The final model was the simplest one that generalized across subjects–a five-stage model. Then, we allowed one stage to have different gamma-scale parameters across experimental conditions, and we used leave-one-subject-out cross-validation to decide on the best model. As in previous studies [4,15], a model with different gamma distributions in the retrieval stage explained the MEG data best.

## Measurements of neural coordination

To measure neural coordination–local synchrony and directed functional connectivity–we used the analytic signal of theta band oscillations. The parcellated MEG data were band pass filtered (cut-off frequencies: 3.8, and 8.5 Hz; forward-backward IIR Butterworth filter of order 4) and epoched from -0.4 seconds before stimulus onset to 0.4 seconds after the response. Epochs were Hilbert transformed to the analytic signal using a symmetric padding of 0.4 seconds to avoid edge artifacts. The analytic signal was transformed into phase and envelope values to compute dpFC and local synchrony, respectively.

Directed functional connectivity between regions $i$ and $j$ was measured with the directed phase-lag index (dpFC) [32] as follows:

$$\text{within} - \text{stage}^s \text{dpFC}^{ij} = \frac{1}{N} \sum\nolimits_{n=1}^{N} \frac{1}{T_n^s} \sum\nolimits_{t=b^s+1}^{T_n^s} \text{sgn}\left(\text{Im}(S_t^{ij})\right) \tag{1}$$

$$\text{across} - \text{trials}_t^s \text{dpFC}^{ij} = \frac{1}{N} \sum_{n=1}^{N} \text{sgn}\left(\text{Im}(S_{nt}^{ij})\right) \tag{2}$$

In Eqs 1 and 2 $Im(S^{ij})$ is the imaginary part of the cross-spectral density between regions $i$ and $j$, $sgn$ is the sign function, N is the number of trials, and $n$ is each individual trial. Then to compute within-stage dpFC at stage $s$ (Eq 1), $b^s +1$ is the first sample after the bump at the onset of the stage $s$, and $T_n^s$ is the number of samples of the flat of the stage $s$ in trial $n$. Therefore, $S_t^{ij}$ is the cross-spectral density over the time points $t$ of the flat in stage $s$ of trial $n$. Across-trials dpFC (Eq 2) was computed at every individual sample $t$ of stage $s$ time-locked to the onset of the very same stage $s$. Therefore $S_{nt}^{ij}$ is the cross-spectral density at sample $t$ of stage $s$ over the $N$ trials of a subject, and nt is the trial index. Both within-stage and across-trial dpFC were later averaged across subjects.

## Generative whole-brain model

The generative whole-brain model (GWBM) was derived with the Ott-Antonsen ansatz [55] from a network-of-networks of Kuramoto oscillators [36]. See [17] for a step-by-step derivation. The dynamics of synchrony in a region are given by the Kuramoto order parameter (KOP) which describes the dynamics of synchrony in in biological systems as well as a pool of neurons [56]. The KOP is a complex number (KOP = $re^{i\psi}$) with the modulus bound by zero

(asynchrony) and one (full synchrony). Here, the KOP simulated the analytic signal of the MEG sources. MEG sensors record the aggregated neural activity generated by millions of neurons in a cortical region. Each active neuron in this region generates an electric field. The measurable electric potential depends on the alignment of active neurons and the temporal synchrony of the dipole moments generated by the electric fields. The neurons that contribute to the MEG signal are parallel to each other. Therefore, the strength of the measurable electric potential in the region is proportional to the synchrony of dipole moments [57,58]. In our model, the KOP represents the synchrony of dipole moments that generate the MEG signals.

Beforehand we set the natural frequencies of the oscillators to a Lorentzian distribution centered in the theta band (center, $\Omega$: 6 Hz, spread, $\Delta$: 1), and the spike-propagation velocity along the structural fibers to 5 m/s. The equations of the KOP in on region, *i*, of the GWBM are as follows:

$$\dot{r}_i = -\Delta_i r_i + \frac{L_i}{2}\left(1 - r_i^2\right)r_i + \frac{G}{2R}\left(1 - r_i^2\right)\sum_{j=1,j\neq i}^{R} A_{ij} r_j\left(t - \tau_{ij}\right)\cos\left(\psi_j(t - \tau_{ij}) - \psi_i\right) \quad (3)$$

$$\dot{\psi}_i = \Omega_i + \frac{G}{2R}\left(r_i + \frac{1}{r_i}\right)\sum_{j=1,j\neq i}^{R} A_{ij} r_j\left(t - \tau_{ij}\right)\sin\left(\psi_j(t - \tau_{ij}) - \psi_i\right) \quad (4)$$

The time dependency has been removed in variables without time delays; $\tau$ are the time delays between regions (fiber length x spike-propagation velocity); *A* is the coupling strength between regions (density of structural fibers); and *R* is the number of regions. To simulate resting state dynamics we explored parameters *G* (global scaling of structural connectivity) and *L* (local connectivity strength, same in all regions) with 25 randomly initialized models. The results of this exploration are shown in S1 Fig. With *G* and *L* set to correctly reproducing the resting state, the thalamocortical inputs were simulated as 0.03 second increases/decreases of *L* at each region and stage onset independently. Simulated dpFC was measured with Eq 1 (within-stage dpFC), but here *N* represented 25 models with different initial conditions and $T_n$ was the median duration of the MEG stages. The initial conditions for the first stage were the MEG phases and amplitudes at the pre-encoding stage plus random noise. More details of the simulations are reported in the SI.

## Generative whole-brain model: Resting-state

To identify a GWBM that simulated resting-state dynamics we performed a grid-search over the global and local coupling parameter space. The local couplings were assumed to be identical for all regions. Resting-state dynamics are characterized by temporal fluctuations of global and local synchrony, and time-resolved patterns of functional connectivity (i.e. metastable dynamics). Metastability was measured as the standard deviation of the modulus of the KOP over time at local and global levels [59,60]. At the local level, the metastabilities were averaged across regions. To obtain the global KOP over time we averaged the phases of the local KOPs across regions ($\psi$ in Eq 3 of the main text). To assess the temporal structure of the global metastability we computed the mean of the absolute values of its autocorrelation function. To avoid the influence of the initial conditions on the simulations we ran twenty-five GWBMs with random initial conditions for each combination of parameters. The simulations were run for 1000 seconds, but the initial 200 seconds were removed to discard initial transients. All simulations were performed with a time-delayed first-order Euler method and an integration step of 1 millisecond. We ended up with a global coupling of 0.15 and a local coupling of 0.7, which had the best trade-off between high metastability and low autocorrelation of global KOP, and therefore were chosen as the default values for the following GWBMs. S1 Fig shows the grid search results of resting-state dynamics.

## Generative whole-brain model: Cognitive task

To simulate the sequence of cognitive stages and their associated neural coordination patterns, we initialized 25 models informed by the theta-band phases and envelope amplitudes observed at the pre-encoding stage. The MEG envelopes were measured 0.1 and 0.05 seconds before the onset of the encoding stage. Then, the initial history of the KOP modulus was a straight line that joined the mean of these amplitudes across trials plus Gaussian noise ($\sigma = 0.01$). To choose the initial history of phases we measured inter-trial phase consistency, and within-stage dpFC at the pre-encoding stage. There were 10 regions (mostly occipital and parietal) that showed significant inter-trial phase consistency. The initial history of phases at these regions were set to the average MEG phases across trials at 0.05 seconds before the onset of the first stage plus Gaussian noise ($\sigma = 0.01$). The phases of these regions were used as a referent point for the remaining regions. The initial phases of the remaining regions were set by an optimization algorithm (CMAES [61]) which tried to establish a phase-lag relationship between regions as in the empirical within-stage dpFC. The dpFC of the initial history of phases had an average similarity to empirical within-stage dpFC of 78%. The 25 GWBMs of the later stages were initialized with the last simulated samples of the previous stage in the best individual of the optimization process (see section *Optimization of thalamocortical inputs*).

The model with the best fitting sequence of parameters was left to run 400 seconds after the last stage. S2 Fig shows that the model neither remained trapped into the functional connectivity state of the last stage, nor did it return to any of the previous states (S2 Fig, bottom). Instead, the model returned to resting state patterns of global and local synchrony for which the functional connectivity fluctuated over time (i.e. metastable dynamics; S2 Fig, top & middle).

## Optimization of thalamocortical inputs

To find the optimal thalamocortical inputs for reproducing the observed connectivity patterns, we used the generalized island model for evolutionary optimization [44]–algorithm DE1220 as implemented in the pagmo toolbox [62]. The generalized island model optimized in parallel ten islands connected in a ring. Each island consisted of 50 individuals and had a particular parametrization of a differential evolution algorithm (see *S1 Table*). The islands occasionally exchanged their best-fitted individuals. This configuration allowed for simultaneously exploring and exploiting multiple areas of the parameter space. Their fitness function had three objectives that were combined into one index of fitness. The dominant objective was to maximize the similarity of simulated and empirical within-stage dpFC, $f_1$:

$$f_1 = \sum_{i=1}^{E} x_i \cdot y_i \left( \sum_{i=1}^{E} |x_i| \right)^{-1} \tag{5}$$

The links, *E*, in the empirical dpFC, *x*, were either 0 (not significant), 1 (lag-ahead) or -1 (lag-behind). Simulated dpFC links, *y*, were either -1 or 1. The objective $f_1$ gave discrete values which interval was used by the other two objectives. The second objective, $f_2$, maximized the topological similarity of the relative change in envelope amplitude at the onset of each stage. This similarity was measured with the Spearman rank-correlation between MEG and simulated relative amplitudes. The third objective, $f_3$, minimized the absolute size of the thalamic pulses as

$$f_3 = \frac{1}{L_{max}} \sum_{j=1}^{R} |L_j| \left( \sum_{i=1}^{E} |x_i| \right)^{-1} \tag{6}$$

where $L_i$ are the local connectivities, and $L_{max}$ is the largest absolute pulse allowed to the

optimizers. The combined fitness index was $f = f_1 - (1-f_2)f_3$. The best individual had the minimal $(1-f_2)f_3$ among the 5000 individuals with the highest $f$ in order to avoid a GWBM with low $f_2$ and $f_3$. The last simulated samples of this individual were used to initialize the simulations of the next stage (see section *Generative whole-brain model*: *Cognitive task*).

S3 Fig shows the parameters of the individuals and their fitness along the evolution in one island as example. This figure shows how the cost function could simultaneously maximize the fitness of within-stage dpFC and relative local synchrony at the onset (Spearman correlation), while the change in local coupling was minimized. The optimization of the four stages took approximately 4 days using 10 CPUs, one for each island.

## Relevance of individual regions for switches

To assess the relevance of a region for switching between states of dpFC, a GWBM was lesioned by setting the thalamocortical pulse in this region to zero while the remaining regions were left untouched. Then, the fitness of the lesioned GWBM (Eq 5) was compared to the fitness achieved by the original GWBM. The relevance of a region was measured as the number of within-stage dpFC links in the lesioned model that were not matching MEG data relative to the number of links matching MEG data in the full model. This process for measuring relevance was repeated for the 68 regions in the GWBM and the four transitions between stages. To obtain a measure of relevance that was not dependent on a single GWBM, relevance was evaluated in 1,000 GWBMs randomly picked from among the models in the top one percentile after optimization. Next, we used linear regression models with one independent variable to explain the relevance of regions. Each linear model included as dependent variable the relevance of the 68 regions and four stages in a lesioned GWBM. A linear model was fitted for each of the 1,000 lesioned models independently.

## Structural connectivity, MRI acquisition and processing

The density and the length of the neural fibers that anatomically connect cortical region was obtained from 45 subjects in the test-retest dataset of the Human Connectome Project (HCP) 3T. This data set consisted of T1-weighted and multi-shell diffusion MRI data. T1-weighted data were acquired with 0.7 mm isotropic voxel size, TE = 2.14 ms, and TR = 2400 ms. Diffusion MRI data were acquired with a 1.25-mm isotropic voxel size, TE = 89.5 ms, and TR 5520 ms, with three shells with b = 1000, 2000, and 3000 s/mm$^2$, each shell with 90 diffusion weighted volumes and 6 non-weighted images [63]. The diffusion MRI data was already preprocessed as described in [64]. In short, diffusion MRI data were corrected for head motion and geometrical distortions arising from eddy currents and susceptibility artifacts [65]. Finally, the diffusion MRI images were aligned to the structural T1 image. The T1w image was parcellated using the Desikan–Killany parcellation [27], resulting in 68 cortical ROIs. Using the T1w image, the probability maps of the different tissues were obtained to create the five tissue-type files [66,67].

Tractography was carried out with constrained spherical deconvolution [68,69]. A multi-tissue response function was calculated [70] and the average response functions were calculated. The multi-tissue fiber orientation distribution was calculated [71] with the average response function ($L_{max}$ = 8). The fiber orientation distribution images had a joint bias field correction and a multi-tissue informed log-domain intensity normalization [72]. Then, tractography was performed with the iFOD2 algorithm [73] using anatomically constrained tractography [74]; generating 10 million streamlines (cutoff at 0.05, default); and using backtracking [74] and a dynamic seeding [75]. The length of the fibers was set to a minimum of 20 mm and

a maximum of 250 mm [74]. To be able to quantitatively assess the connectivity, SIFT2 was applied to the resulting tractograms [75].

The connectivity matrix was built with a robust approach. In particular a 2-mm radial search at the end of the streamline was performed to allow the tracts to reach the gray matter parcellation [76]. Each connectivity matrix was multiplied by its $\mu$ coefficient obtained from the SIFT2 process, as the sum of the streamline weights needs to be proportional to the units of fiber density for each subject [77]. Connectivity matrices were averaged across subjects, and the 10% of links with the highest coefficient of variation across subjects were set to zero [78]. Finally, the averaged and thresholded structural connectivity matrix was normalized to have an average value of one.

## Supporting information

**S1 Table. Parameters of DE1220 algorithm on each island.**
(PDF)

**S1 Fig. Resting state neural coordination dynamics.** The green dot indicates the parametrization of the model. The location of the green dot was based on the idea that resting state dynamics should have simultaneously the lightest color in the three panels and the weakest coupling parameters.
(PDF)

**S2 Fig. Return to resting-state after cognitive stages.** (top) Modulus of the global KOP. (middle) Modulus of the local Kuramoto order parameter (KOP) for the cortical 68 ROIs. (bottom) Temporal evolution of simulated-to-MEG fitness of *within-stage dpFC* for the four cognitive stages. This is similar to Fig 2B but for a much longer period of time. The MEG *within-stage dpFC* of each stage (Fig 1B) were compared (Eq 5) with the simulated *dpFC* sample-by-sample (Eq 2).
(PDF)

**S3 Fig. Individuals and their fitness along the optimization in one island.** (A) fitness index *f*. (B) Spearman correlation, objective $f_1$. (C) Sum of the absolute change in local coupling at the onset of the stage. Blue dots are the A, B and C values in the order that they were evaluated along the optimization process. Orange dots are the same values but sorted by the Fit Index (A). (D) Change in local coupling (thalamic input) at the onset that produces the blue dots in A, B, C. (E) Same as (D) but sorted by their Fit Index.
(PDF)

**S4 Fig. Mean and standard error of the mean of the relevance of each region for switching between two states of within-stage dpFC.** Average values are obtained for the 1,000 GWBM randomly picked from the 10,000 GWBMs with the best fitness index.
(PDF)

## Acknowledgments

HCP datasets were provided by the Human Connectome Project, WU-Minn Consortium (Principal Investigators: David Van Essen and Kamil Ugurbil; 1U54MH091657) funded by the 16 NIH Institutes and Centers that support the NIH Blueprint for Neuroscience Research; and by the McDonnell Center for Systems Neuroscience at Washington University. We thank Lionel Newman from the University of Groningen for proofreading the manuscript.

## Author Contributions

**Conceptualization:** Oscar Portoles.

**Data curation:** Oscar Portoles, Manuel Blesa, Jelmer P. Borst.

**Formal analysis:** Oscar Portoles.

**Methodology:** Oscar Portoles.

**Software:** Oscar Portoles.

**Supervision:** Jelmer P. Borst.

**Visualization:** Oscar Portoles.

**Writing – original draft:** Oscar Portoles, Jelmer P. Borst.

**Writing – review & editing:** Manuel Blesa, Marieke van Vugt, Ming Cao, Jelmer P. Borst.

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
