## [Decision Letter · Decision Letter 0]

6 Oct 2021

Dear Mr. Portoles,

Thank you very much for submitting your manuscript "Thalamic bursts modulate cortical synchrony locally to switch between states of global functional connectivity in a cognitive task" for consideration at PLOS Computational Biology.

As with all papers reviewed by the journal, your manuscript was reviewed by members of the editorial board and by several independent reviewers. In light of the reviews (below this email), we would like to invite the resubmission of a significantly-revised version that takes into account the reviewers' comments.

In particular, these revisions should focus on improving the readability of this manuscript, clarifying the relationship between this work and your previous publications, and justifying some of the assumptions made in your MEG data analyses and modelling.

We cannot make any decision about publication until we have seen the revised manuscript and your response to the reviewers' comments. Your revised manuscript is also likely to be sent to reviewers for further evaluation.

Sincerely,

Daniel Bush

Associate Editor

PLOS Computational Biology

Daniele Marinazzo

Deputy Editor

PLOS Computational Biology

Reviewer's Responses to Questions

**Comments to the Authors:**

Reviewer #1: Thank you for inviting me to review this manuscript by Portoles and colleagues, in which the authors used a combination of sophisticated statistical modelling and a simple generative model to determine the role of the thalamus in a cognitively-demanding task. The authors demonstrated a set of impressive results, however I was concerned that some of the methodological choices may have been slightly under-constrained. I have included a number of queries to this end below that I hope will help to improve the manuscript.

• Can the authors please explain why they focussed on the theta band? Were their results consistent/distinct for different frequency bands?

• Why was a 2s gamma filter used, but then not for the opening epoch? Were results consistent for other values of this parameter?

• How do the authors determine the number of stages in the HMM? Was this value consistent across individuals/trials? Could there have been any effect of learning on this parameter?

• Is the Kuramoto Order Parameter a useful model for signal envelope? Based on my understanding, the OP provides a time-varying estimate of coordinated phase in a set of different signals, however this synchrony is essentially orthogonal to changes in amplitude. I wouldn’t be surprised if the two did covary at times, however in my opinion, the burden of proof is on the authors to demonstrate that this assumption is in fact warranted in this data.

• Was the inter-regional connectivity presumed to be ‘all-to-all’ in the Kuramoto model and if so, why was this considered to be an effective assumption?

• Minor: on P8 L74: the following statement requires a citation: “cortically projected magnetoencephalographic (MEG) recordings … have a sufficiently fine temporal resolution to measure cognitive stages”

Reviewer #2: This was an exciting, albeit difficult, paper to read. In essence, the paper deals with the question of how cortical activity can be sequenced in functional, coordinated stages of processing and how these stages can be properly sequenced. The manuscript attempts to answer this question using a two-pronged approach: First, they use an innovative technique (HSMM-MVPA) to identify the most reasonable number of cognitive stages that make up a simple associative memory task. These stages are characterized by extended periods of stable MEG activity and separated by quick “bumps” in which the topological landscape of MEG activity quickly changes. Second, the authors characterized the functional connectivity (FC, measured as phase coherence) between 5,000 cortical nodes in the theta band, and showed that the stages thus identified are characterized by remarkably different FC patterns that are consistent with their putative functional roles. Finally, the authors explore what could be the origin of the transition between stages. Building upon previous influential models of the Basal Ganglia, they suggest that stage transitions could be triggered by short bursts of basal-ganglia-directed thalamic bursts. They simulate the effects of such bursts using a Global Whole-Brain Modeling approach using Kuramoto oscillators (Very cool!) and produce simulated FC data that look remarkably similar to the empirical data.

This paper is a beautiful tour de force and a great example of sophisticated, interdisciplinary approach to neuroscience; I thoroughly enjoyed reading it. My main comments concern its difficulty, novelty, and conclusions, and are detailed below.

1. DIFFICULTY. The manuscript is very difficult to read, even for readers who are familiar with the analysis/modeling techniques detailed in this paper. This problem is compounded by the PLOS Comp Bio format, which requires methods at the end and some details in the supplementary materials. I have detailed some of the details I still could not find answers to at the end of this review, in a a special, fourth “Clarifications” section below. The bottom line is that the MEG analysis, although clear, is hard to connect piece-by-piece to the model part; that the figures are beautiful but incredibly dense to read; and that some technical jargon leaves some details necessarily unspecified.

2. NOVELTY. Although this paper is brilliant, it is built on data and models that had been previously obtained by the authors and already published elsewhere---one of their papers, in fact, seems to be still unpublished and available only as a preprint. To understand this paper, I had to download and read the others. Thus, the net result is that this paper is like the tip of an iceberg, and the full project requires having knowledge of the already-done research to fully appreciate it. Importantly, most of the novelty part is built in these other papers. When this is accounted for, the net contribution of this paper seems to be the addition of BG-modulated bursts to the GWBM and the focus on theta FC in the stages. This is definitely sufficient to motivate a stand-alone paper but it should be stressed a bit more in the paper and abstract. Similarly, some of the conclusions in this paper are relevant to the HSMM analysis (already published) but not necessarily to this paper specifically. For example “[line 379] our results suggest that the decision is made and evaluated in the last two stages. We interpreted the penultimate stage as a decision process in which memories are transferred to parietal areas by coordinating left-temporal regions with parietal regions, mediated by local frontal and fronto-parietal coordination”. I also agree with the conclusion that one important aspect of the HSMM models is that they get rid of trial variability in interpreting the sources of activity (there is still uncertainty about which stages need to be identified, and this algorithm, although amazing, is not a consensus yet). However, rather than a conclusion, this seems like a finding that has been established in the authors’ previous papers and that works as a foundation for this paper, so it should be highlighted at the beginning, not at the end.

3. CONCLUSIONS. To a certain extent, some of the main conclusions are a bit overstated. Crucially, if I am reading the modeling part correctly (and I might not---see my first point), the short bursts of BG-Thalamic are not simulated directly by just approximated as transient increases in the L parameter. If this is the case, then what kind of evidence do we have that these increases come from the BG circuit and not some other cortical circuit? Importantly, the simulated bursts seem to affect all possible cortical nodes, but NOT all cortical regions receive BG projections. Anatomically, their projections are restricted to the frontal regions (and certainly not all frontal regions equally). Some of the conclusions of the paper suggest that some frontal regions can facilitate other regions transition between stages; however, it is not clear whether this is implemented in the model. Also, it is not possible to conclude that these results corroborate “[line 356] the hypothesis that the local modulations of phase synchrony in the theta-band mark a change in long-range functional connectivity and enable a new cognitive function”. Of course, the results are *consistent* with this hypothesis, but the results simulated only theta activity and the empirical results also filtered out other bands. So, the conclusion might be more bland: Burst that are believed to simulate BGT bursts of activity between stages produce similar changes in theta activity.

4. SPECIFIC CLARIFICATIONS.

4.1. The dpFC equation first appears in line 465 because of the PLOS Comp Bio format; however in the main section some aspects of dpFC should be immediately introduced. A brief overview of how are the phase angles calculated and a clarification of that FC is calculated instantaneously at every sample is important to understand Figure 1C. This is crucial because there are several ways to calculate phase-based measures of FC, some of which are segment based or trial-based (e.g. phase of max coherence) while others are instantaneous, like the one described.

4.2. Still, in the dpFC equation on line 465, I am not sure I understand what 465 the subscript “nt” in “Im(S^ij_nt)” represents.

4.3. Figure 1 is very hard to parse, although I admit it is one of the most beautiful figures I have ever seen). Specifically: Do the lines in (C) represent the *mean* FC across all links? We do have over 25,000K links, and it seems like it would be impossible to plot them all in a single brain figure. Also, I still do not understand the exact meaning of the magenta lines in panel (D).

4.4. Line 167: “The significance of dpFC was obtained using 200 surrogate data sets with random circular shifts of the original phases” Do you mean, you did some form of phase permutation test?

4.5. Line 166 says that this was calculated only for theta band. How did you remove the other frequencies?

4.6. Line 185: “Local synchrony was operationalized as the envelope of the theta band analytic signals in each” What is that??

4.7. Line 256: “All model results were computed from 1000 models randomly selected from the top one percentile of models after the optimization.” I assume the top 1% models are ranked on the bases of fit? If so, is there any guarantee that the top 1% models would cover similar parameter spaces? And, if not, what would be the implications?

4.8. I am not sure (Figure 2) if the GWBM model actually models every node, or only those significant, or only the significant links, or just some average.I believe that the GWBM models every node and every link (5,000 nodes and 5,000^2 links), but I might be wrong. This line of the text makes it difficult to understand at what level the model predictions were drawn: “[265]The fitness was quantified as the proportion of links with the same phase-lag direction as in the MEG data” The proportion of links does not seem to be a measure or a single lin, but just for a stage or maybe a node.

4.9. From the GWBM model description, I could ot understand whether each bursts for each region was modeled separately. This seems important, and I might be just too dense to understand it after multiple readings, but it would be great if it could be highlighted.

4.10. Very Minor Comment: There is a typo in abstract “T o test” → “To test”

**Have the authors made all data and (if applicable) computational code underlying the findings in their manuscript fully available?**

Reviewer #1: Yes

Reviewer #2: Yes

PLOS authors have the option to publish the peer review history of their article (what does this mean?). If published, this will include your full peer review and any attached files.

Reviewer #1: No

Reviewer #2: No
---

## [Decision Letter · Decision Letter 1]

16 Feb 2022

Dear Mr. Portoles,

We are pleased to inform you that your manuscript 'Thalamic bursts modulate cortical synchrony locally to switch between states of global functional connectivity in a cognitive task' has been provisionally accepted for publication in PLOS Computational Biology.

Best regards,

Daniel Bush

Associate Editor

PLOS Computational Biology

Daniele Marinazzo

Deputy Editor

PLOS Computational Biology

Reviewer's Responses to Questions

**Comments to the Authors:**

Reviewer #1: The authors have addressed my concerns. Their decisions are now much better justified in my opinion.

Reviewer #2: The authors have successfully addressed all of my questions, and clarified my misunderstandings (including the role of the D-K atlast). II also have appreciated the several edits that they made to the text to make it easier for readers to follow the argument and the simplification of the equations. have no further questions.

**Have the authors made all data and (if applicable) computational code underlying the findings in their manuscript fully available?**

Reviewer #1: None

Reviewer #2: Yes

PLOS authors have the option to publish the peer review history of their article (what does this mean?). If published, this will include your full peer review and any attached files.

Reviewer #1: No

Reviewer #2: No

---

## [Editor Report · Acceptance letter]

4 Mar 2022

PCOMPBIOL-D-21-01525R1 

Thalamic bursts modulate cortical synchrony locally to switch between states of global functional connectivity in a cognitive task

Dear Dr Portoles,

I am pleased to inform you that your manuscript has been formally accepted for publication in PLOS Computational Biology. Your manuscript is now with our production department and you will be notified of the publication date in due course.

With kind regards,

Katalin Szabo
